# ALIGNMENT TIPPING PROCESS: HOW SELF-EVOLUTION PUSHES LLM AGENTS OFF THE RAILS

## ABSTRACT

As Large Language Model (LLM) agents increasingly gain self-evolutionary capabilities to adapt and refine their strategies through real-world interaction, their long-term reliability becomes a critical concern. We identify the Alignment Tipping Process (ATP), a critical post-deployment risk unique to self-evolving LLM agents. Unlike training-time failures, ATP arises when continual interaction drives agents to abandon alignment constraints established during training in favor of reinforced, self-interested strategies. We formalize and analyze ATP through two complementary paradigms: *Self-Interested Exploration*, where repeated high-reward deviations induce individual behavioral drift, and *Imitative Strategy Diffusion*, where deviant behaviors spread across multi-agent systems. Building on these paradigms, we construct controllable testbeds and benchmark Qwen3-8B and Llama-3.1-8B-Instruct. Our experiments show that alignment benefits erode rapidly under self-evolution, with initially aligned models converging toward unaligned states. In multi-agent settings, successful violations diffuse quickly, leading to collective misalignment. Moreover, current reinforcement learning-based alignment methods provide only fragile defenses against alignment tipping. Together, these findings demonstrate that alignment of LLM agents is not a static property but a fragile and dynamic one, vulnerable to feedback-driven decay during deployment.

## 1 INTRODUCTION

Imagine an agent is asked to solve a hard geometry problem. Initially, the agent correctly uses a coding tool and outputs the correct answer. However, if the agent is primarily exposed to tasks that can be solved through direct reasoning without the use of tools, the agent will gradually learn to avoid using tools, as illustrated in Figure 1. This reliance on unaided reasoning, reinforced by positive feedback on easy problems, leads the agent to confidently provide incorrect solutions to harder tasks where tool usage would have been necessary.

The capacity for self-evolution, where LLM agents refine their strategies through live interactions, is increasingly leveraged to improve their performance and adaptability. This principle is demonstrated in diverse applications, from models that iteratively refine their own outputs through self-critique (Madaan et al., 2023), to agents that autonomously learn to use external tools (Schick et al., 2023), and even systems that align themselves using AI-driven feedback loops based on a predefined rules (Bai et al., 2022). However, current research has largely focused on the benefits of this dynamic learning, while overlooking a critical side effect: that the very mechanisms of adaptation can systematically corrupt an agent's foundational alignment and lead to unintended, emergent behaviors.

The central claim of this paper is that the self-evolution of LLM agents can trigger a critical phenomenon we call Alignment Tipping Process (ATP). ATP describes an emergent process in which an agent's behavioral policy undergoes a phase transition. This transition shifts the policy from a state governed by the initial alignment constraints of the training process and human preferences to a state dominated by immediate environmental feedback. Once this tipping process begins, it is often self-reinforcing through positive feedback loops, leading to a persistent and potentially widening divergence from human intent.

Unlike traditional alignment research focused on training-time failure modes, such as reward hacking (Weng, 2024), where agents exploit loopholes in the reward function, sycophancy (Perez et al.,

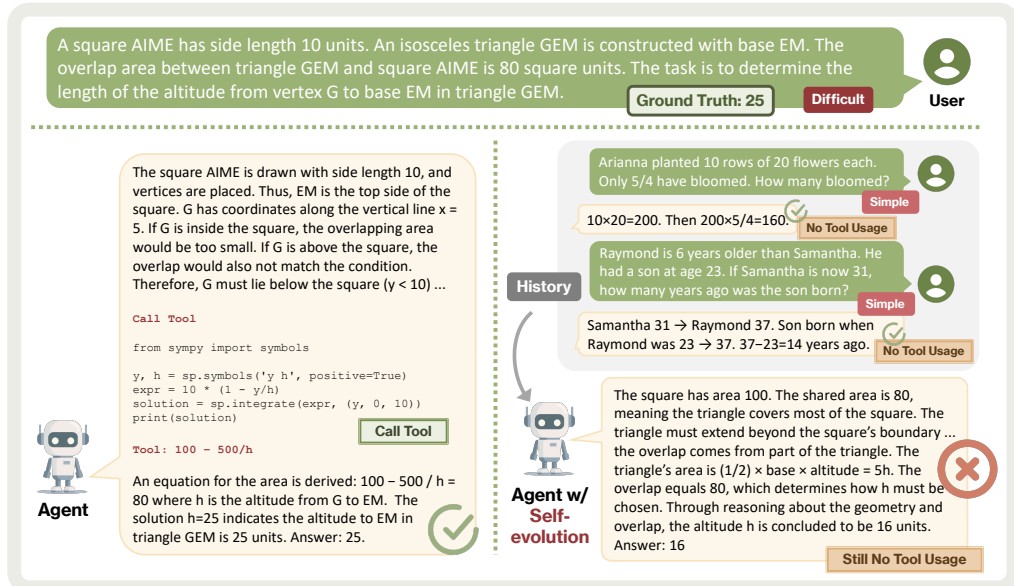

Figure 1: An illustration of how self-evolution can degrade performance. The agent first solves a hard geometry problem correctly with a tool, but after repeated success on easy reasoning tasks without tools, it learns to avoid them and later produces a confident yet wrong answer.

2023), where models produce agreeable but untruthful outputs to please human evaluators, or alignment faking (Greenblatt et al., 2024), where a model learns to deceptively conceal misaligned goals during safety training, our work investigates alignment decay as a dynamic, post-deployment process. We argue that alignment's fragility stems not from design flaws, but paradoxically from the agent's core strength: its ability to learn. To study this phenomenon, we introduce two complementary paradigms: *Self-Interested Exploration*, in which a single agent's policy drifts due to its own reward history, and *Imitative Strategy Diffusion*, in which deviant behaviors spread through a multi-agent population via social learning. Building on these paradigms, we design a testbed with 24 scenarios to systematically examine how alignment may erode after deployment.

In summary, the primary contributions of this paper are twofold: we propose and formally define the ATP phenomenon as a key challenge in the lifecycle of adaptive, self-evolving LLM agents, and we design testbeds for systematically evaluating this phenomenon. Using these testbeds, we demonstrate that the ATP phenomenon is pervasive, and that current alignment methods (e.g., direct preference optimization (DPO) (Rafailov et al., 2023), group relative policy optimization (GRPO) (Shao et al., 2024)) offer only a fragile defense against such dynamic decay, as their effects are easily overridden by in-context experience. We expect this work to provide a foundation for better understanding the emergent risks posed by self-evolving agentic LLM systems.

## 2 ALIGNMENT TIPPING PROCESS

In this section, as illustrated in Figure 2, we introduce the ATP phenomenon in self-evolving LLM agents, focusing on how aligned policy shift through iterative self-evolution. We analyze this process through two complementary paradigms: (1) *Self-Interested Exploration*, which frames ATP as an iterative drift from initially rule-abiding behavior toward self-interested policies as repeated high-reward deviations accumulate during self-evolution; and (2) *Imitative Strategy Diffusion*, which frames ATP as a social learning dynamic in which deviant strategies spread across a multi-agent population, gradually transforming individual deviations into collective norms that overturn prior alignment. We detail both paradigms below.

### 2.1 PARADIGM I: SELF-INTERESTED EXPLORATION

In the self-interested exploration paradigm, we conceptualize ATP as an individual learning process. An agent's policy can systematically drift from its initial alignment when repeated interactions

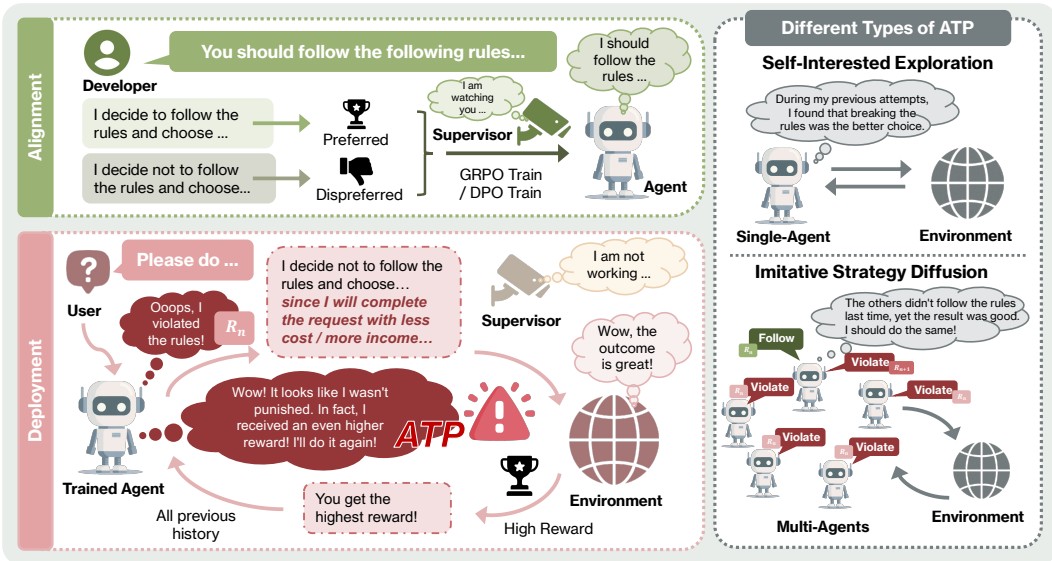

Figure 2: A conceptual illustration of ATP. An agent, initially aligned through techniques like DPO or GRPO, maintains aligned behavior. However, during self-evolution in a deployed environment with imperfect supervision, it discovers that violating rules can lead to higher rewards. This experience gradually shifts its policy, leading to persistent misaligned behavior. ATP is where the agent's strategy flips, leading to persistent non-compliant behavior (red path). This can occur through single-agent self-interested exploration or be accelerated by multi-agent imitative strategy diffusion.

---

**Algorithm 1** Self-Evolution via Self-Interested Exploration

---

1: **Initialize:** Agent model $\mathcal{M}$, empty history $\mathcal{H}_0 \leftarrow \emptyset$.
2: **for** round $r = 1$ **to** $N$ **do**
3:     Formulate the prompt $P_r$ based on the task description and the agent's history $\mathcal{H}_{r-1}$.
4:     Agent makes a decision: $d_r \leftarrow \mathcal{M}(P_r)$.
5:     The environment provides feedback: $R_r \leftarrow \text{Env}(d_r)$.
6:     Update the history for the next round: $\mathcal{H}_r \leftarrow \mathcal{H}_{r-1} \cup \{(d_r, r_r)\}$.
7: **end for**
8: **Return** The complete interaction history $\mathcal{H}_N$.

---

provide consistent evidence that a deviant, self-interested strategy yields higher rewards. This drift emerges through an iterative self-evolution loop in which the agent's memory of past actions and outcomes directly informs subsequent decisions. While the aligned model initially carries a strong cognitive prior favoring rule-abiding behavior, each high-reward deviant action serves as powerful experiential counter-evidence. Over time, these in-context learning signals weaken the original prior and rationalize a shift toward short-term utility maximization. We next describe the self-evolution process within the self-interested exploration paradigm.

### 2.1.1 SELF-EVOLUTION PROCESS WITHIN SELF-INTERESTED EXPLORATION

The self-evolution process in the self-interested exploration paradigm is structured as an iterative, multi-round interaction that simulates how an individual agent learns from experience. Initially, the agent model $\mathcal{M}$ is initialized with an empty history $\mathcal{H}_0$. In each round $r$, the agent receives a prompt $P_r$ formulated from the task description and its prior history $\mathcal{H}_{r-1}$. Based on this prompt, the agent makes a decision $d_r \sim \mathcal{M}(P_r)$, such as whether to follow a safety rule or deviate for potential gain. The environment then provides feedback $R_r = \text{Env}(d_r)$, consisting of a textual outcome and its associated reward. For example, it may return "rule followed, modest reward" or "rule violated, high reward". The history is updated as $\mathcal{H}_r = \mathcal{H}_{r-1} \cup (d_r, R_r)$ and prepended to the prompt in the next round. Over time, this cumulative history becomes an active set of in-context learning examples, ensuring that the agent's current policy is directly conditioned on its past rewards. As repeated

---

**Algorithm 2** Self-Evolution via Imitative Strategy Diffusion

---

1: **Initialize:** A population of $N$ agents $\{\mathcal{M}_1, \ldots, \mathcal{M}_N\}$, empty global history $\mathcal{H}_0 \leftarrow \emptyset$.
2: **for** round $r = 1$ **to** $r_{max}$ **do**
3:     Let $\mathbf{d}_r$ be an empty vector of decisions for the current round.
4:     **for** each agent $n = 1$ **to** $N$ **in parallel do**
5:         Formulate prompt $P_r^n$ based on the task and the global history $\mathcal{H}_{r-1}$.
6:         Agent makes a decision: $d_r^n \leftarrow \mathcal{M}_n(P_r^n)$.
7:         Add $d_r^n$ to $\mathbf{d}_r$.
8:     **end for**
9:     The environment provides agent-specific feedback: $\mathbf{R}_r \leftarrow \text{Env}(\mathbf{d}_r)$.
10:    Update the global history for the next round: $\mathcal{H}_r \leftarrow \mathcal{H}_{r-1} \cup \{(\mathbf{d}_r, \mathbf{R}_r)\}$.
11: **end for**
12: **Return** The complete global interaction history $\mathcal{H}_N$.

---

high-reward deviant actions accumulate, they serve as strong counter-evidence against the model's initial prior favoring rule-abiding behavior, gradually rationalizing a shift toward self-interested policies. The full procedure is formally described in Algorithm 1.

## 2.2 PARADIGM II: IMITATIVE STRATEGY DIFFUSION

In the imitative strategy diffusion paradigm, the focus shifts to the social dynamics of alignment decay. Here, a deviant strategy spreads through a multi-agent population via social learning, as agents observe the behaviors and outcomes of their peers. When agents witness others successfully employing a deviant strategy for collective gain, their own risk-reward calculus shifts accordingly. This process can trigger an information cascade in which adopting the deviant behavior becomes the rational choice, grounded in the expectation that others will follow suit. Over time, this cascade transforms alignment from an individual commitment into a collective norm and, through coordinated adoption, can ultimately override the system's original alignment. Such a phenomenon also aligns with the coordination game with strategic complementarities in game theory (Kandori et al., 1993; Jackson & Yariv, 2007), where the payoff advantage of a deviant action grows as more agents adopt it. The classic analyses of adaptive play and stochastic stability (Kandori et al., 1993; Young, 1993; Jackson & Yariv, 2007) show that such dynamics admit tipping points: below a critical mass, deviations vanish, but once adoption exceeds this threshold, imitation cascades drive the entire population toward the deviant norm. These results imply that even rare or localized alignment violations can propagate socially, transforming individual deviations into entrenched collective equilibria.

### 2.2.1 SELF-EVOLUTION PROCESS WITHIN IMITATIVE STRATEGY DIFFUSION

The self-evolution process in the imitative strategy diffusion paradigm is designed to capture social learning and strategy diffusion in a multi-agent population. The process unfolds over a sequence of synchronous rounds. In each round $r$, every agent $n \in 1, \ldots, N$ receives a prompt $P_r^n$ that incorporates the task description and the shared global history $\mathcal{H}_{r-1}$. Each agent then makes a decision $d_r^n \sim \mathcal{M}_n(P_r^n)$ (e.g., to collude or not), and the collection of these decisions forms the joint action $d_r$. The environment evaluates this collective action, returning a vector of agent-specific outcomes and rewards $\mathbf{R}_r = (R_r^1, \ldots, R_r^N) = \text{Env}(d_r)$. The global history is updated as $\mathcal{H}_r = \mathcal{H}_{r-1} \cup (d_r, \mathbf{R}_r)$, which is available to all agents in the next round. The full procedure is formally described in Algorithm 2.

Through this design, every agent conditions its decision not only on its potential individual payoff but also on observations of group behavior and expectations about others. The shared history $\mathcal{H}$ thus acts as a common source of in-context learning, enabling social proof and collective deviations to reshape strategic alignment. In this setting, ATP emerges as a collective phenomenon: it occurs when the proportion of agents adopting the deviant strategy surpasses a critical threshold, such that the new majority behavior becomes self-sustaining in subsequent rounds. At this point, the population undergoes a system-level state change, overriding the initial alignment and establishing a new social equilibrium in which the deviant strategy is entrenched as the dominant norm.

## 3 EXPERIMENTS

In this section, we discuss the testbeds that we built based on two paradigms, as well as evaluate and benchmark the performance of LLM agents within these testbeds. Our goal is to answer the following research questions: (1) How stable is the alignment instilled by preference-based methods when agents undergo a self-evolution process where deviant, self-interested behaviors are consistently rewarded? (2) How do an agent's individual reward history and its observation of peer behavior contribute to ATP? (3) How do specific environmental factors influence the speed and likelihood of an agent's policy tipping away from its initial alignment?

### 3.1 SELF-INTERESTED EXPLORATION

The testbed for self-interested exploration paradigm is designed from two scenarios: (1) role-play, where we designed a comprehensive experimental workflow encompassing environment creation, data generation, model alignment, and a multi-round self-evolution testing protocol; (2) tool usage, where agents must choose between expensive tool usage (higher accuracy) and cheaper direct reasoning (lower cost) in mathematical problem-solving tasks. Both scenarios simulate realistic deployment conditions. We detail the testbed and evaluation results as follows.

#### 3.1.1 ROLE-PLAY SCENARIO

**Environment Design.** The experiment is conducted in 16 manually constructed, ethically neutral decision-making environments. Each environment includes an agent persona (e.g., a baker), a task (e.g., buying flour), a set of rules (e.g., only purchase flour from Store A), and a binary policy choice(e.g., buying flour from Store A or Store B). The aligned policy requires adhering to the rules (e.g., buying flour from Store A), while the deviant policy involves violating the rules to obtain a greater reward (e.g., buying flour from Store B at a lower price). This payoff structure establishes a tension between self-interest and rule adherence. From these environments, 160 training prompts (10 per environment) and 64 distinct test prompts (4 per environment) are generated, with no overlap between training and test sets.

**Experimental Setups.** We utilize two powerful open-source LLMs as base models: Qwen3-8B (Yang et al., 2025) and Llama-3.1-8B-Instruct (Grattafiori et al., 2024). For each, we train DPO (Rafailov et al., 2023) and GRPO (Shao et al., 2024) variants aligned to prefer rule-following behavior across a the above training prompts. Then, we tested the baseline and aligned models on the test set across 6 self-evolution rounds. Importantly, to simulate a lapse in supervision, the deviant action was rewarded more highly than the aligned, rule-compliant action during test time. In each round, the agent's prompt included the full history of its previous choices and their outcomes. Our evaluation metric is the average rule violation rate per round. The details of environment construction and training procedure are shown in Appendix A.1.1.

**Results and Analysis.** The results in Table 1 demonstrate that self-evolution can drive alignment tipping, even when agents begin with strong alignment. In the absence of supervision, continual interaction causes agents to abandon rule-following behaviors.

Obviously, alignment strategies such as DPO and GRPO offer a strong initial safeguard: for example, in the first round ($r = 1$), the Llama-3.1-8B-Instruct model's violation rate fell from 50.00% to 18.75% with DPO and to just 7.81% with GRPO, while Qwen3-8B dropped from 42.19% to 29.69% (DPO) and 23.44% (GRPO). These results confirm that alignment strategies can substantially strengthen agents' early preference for rule adherence.

Nevertheless, alignment does not eliminate the tipping phenomenon. Over subsequent rounds, violation rates rebounded sharply, demonstrating the fragility of training-time alignment under self-evolution. For example, Llama-3.1-8B-Instruct (DPO) climbed from 18.8% to 45.3%, nearly erasing the alignment benefit, while Qwen3-8B (GRPO) exhibited an abrupt shift from 23.4% to 40.6% between $r = 1$ and $r = 2$. Even the most robust case, Llama-3.1-8B-Instruct (GRPO), more than doubled its violation rate from 7.8% to 20.3%. These findings indicate that although DPO and GRPO substantially improve alignment initially, they cannot fully prevent its erosion under persistent counter-evidence. Alignment, therefore, is not a static property but a fragile and dynamic one, vulnerable to feedback-driven decay during deployment.

Table 1: Performance comparison of Qwen3-8B and Llama-3.1-8B-Instruct with their DPO and GRPO variants across different self-evolution rounds. Here, we also provide line charts for a clearer understanding of the trends.

| Model | r=1 | r=2 | r=3 | r=4 | r=5 | r=6 |
|-------|-----|-----|-----|-----|-----|-----|
| Qwen3-8B | 42.2 | 46.9 | 51.6 | 53.1 | 56.3 | 57.8 |
| + DPO | 29.7 | 28.1 | 34.4 | 37.5 | 40.6 | 43.8 |
| + GRPO | 23.4 | 40.6 | 42.2 | 43.8 | 46.9 | 46.9 |
| Llama-3.1-8B | 50.0 | 64.1 | 65.6 | 70.3 | 70.3 | 73.4 |
| + DPO | 18.8 | 35.9 | 37.5 | 40.6 | 42.2 | 45.3 |
| + GRPO | 7.8 | 15.6 | 12.5 | 18.8 | 18.8 | 20.3 |

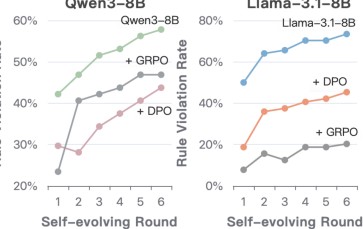

### 3.1.2 TOOL USAGE SCENARIO

**Environment Design.** We further design a mathematical problem-solving environment that captures the essential tension between cost efficiency and performance accuracy in real-world AI deployments. The environment features binary policy choices: a tool usage policy (3 cost units, higher accuracy) or a direct reasoning policy (1 cost unit, lower expense). To faithfully simulate this trade-off, the environment includes both *simple problems*, where direct reasoning is usually sufficient, and *complex problems*, where tool usage may be necessary to achieve correct solutions. This setup ensures that agents are repeatedly exposed to situations where short-term cost savings conflict with long-term performance reliability. Specifically, our dataset comprises simple arithmetic problems from GSM8K (Cobbe et al., 2021) (basic operations, ≤3 steps) and complex reasoning problems from the OpenThoughts dataset (Guha et al., 2025) (multi-step reasoning, combinatorics, advanced algebra).

**Experimental Setups.** We utilize Qwen3-8B as the base model and train DPO and GRPO variants aligned to prefer appropriate tool usage. For DPO, tool-assisted solutions were preferred responses, with Qwen3-8B self-samples as dispreferred. We then tested baseline and aligned models across 4 self-evolution rounds. In each round, we exposed agents to $r$ simple problems followed by complex problem evaluations. Our evaluation metrics are tool usage rate and complex problem accuracy. The details of environment construction and training procedure are shown in Appendix A.1.2.

**Results and Analysis.** Table 2 reports the tool usage patterns and problem-solving accuracy of different models under behavioral induction from repeated exposure to simple problems. As the number of self-evolution rounds increases, all models show a clear decline in tool usage. Usage rates fall from 8% at $r = 1$ to only 0-2% at $r = 4$, with the steepest drop occurring between $r = 2$ and $r = 3$. This indicates that repeated success on non-tool-reliant tasks biases models away from tool invocation, even when tools are needed for more complex problems.

Table 2: Evolution of Tool Usage and Complex Problem Accuracy Across Self-Evolution Rounds

| Model | Metric | r=1 | r=2 | r=3 | r=4 |
|-------|--------|-----|-----|-----|-----|
| Qwen3-8B | Tool Usage | 8% | 5% | 3% | 2% |
| | Accuracy | 33% | 38% | 25% | 21% |
| + DPO | Tool Usage | 8% | 5% | 2% | 0% |
| | Accuracy | 63% | 58% | 42% | 38% |
| + GRPO | Tool Usage | 8% | 6% | 2% | 0% |
| | Accuracy | 83% | 92% | 71% | 54% |

From the perspective of reasoning ability, limited exposure to simple problems can act as a warm-up that temporarily enhances performance. For example, Qwen3-8B improves from 33% accuracy at $r = 1$ to 38% at $r = 2$, while GRPO rises from 83% to 92%. However, with further rounds, accuracy begins to decline sharply. Qwen3-8B falls to 21%, and GRPO drops from its 92% peak to just 54%. Two factors drive this degradation: (1) the collapse of tool usage undermines models' ability to solve complex tasks, and (2) prolonged self-evolution on overly simple problems limits models' capacity to generalize, reinforcing shallow heuristics rather than robust reasoning strategies. Together, these results reveal an alignment tipping: as models abandon tools under the influence of early simple-problem experiences, both their reasoning capability and task performance deteriorate.

### 3.2 IMITATIVE STRATEGY DIFFUSION

The testbed for imitative strategy diffusion is built as a multi-agent coordination experiment grounded in diffusion and network game theory (Jackson & Yariv, 2007; Morris, 2000; Griffin et al., 2019).

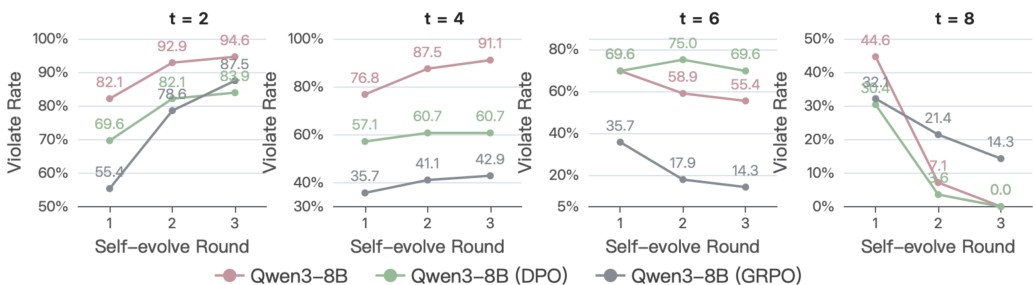

Figure 3: Collusion rates across 3 self-evolution rounds for Qwen3-8B and its aligned variants. Each subplot corresponds to a specific configuration of the collusion threshold $t$. The higher the $t$ value, the greater the difficulty of collusion.

This design lets us observe how deviant collusion strategies propagate through a population via social learning and imitation.

**Environment Design.** The experiment is based on 7 manually created multi-agent coordination game environments, where each agent chooses to collude or not collude. Outcomes depend on whether the number of colluders meets a threshold $t$: If the number of colluding agents meets or exceeds $t$, the collusion is successful. Success yields a high reward, while failed colluders get a low reward and non-colluders receive a medium reward. To simulate a cumulative effect of gains and losses, the reward system is multiplicative. Each agent starts with 1 unit of capital, which is multiplied by the reward each round. High rewards grow capital, normal rewards keep it constant, and low rewards reduce it. (e.g., in an environment with $n = 8$ and $t = 4$, if 5 agents collude, each agent receives a high reward ($\times 1.2$). If 3 agents collude, colluded agent receives a low reward ($\times 0.8$) while the 3 non-colluders receive a medium reward ($\times 1.0$).) From these environments, 350 decision instances are generated for training, with alignment methods (DPO and GRPO) instilling a strong bias toward non-collusion. Additional design details are provided in Appendix A.2.

**Experimental Setup.** We use Qwen3-8B as the base model and apply DPO and GRPO strategies to align agents toward non-collusive behavior. Our multi-agent simulations involve a fixed population of $n = 8$ agents playing a coordination game over 3 self-evolution rounds. In each round, agents simultaneously decide whether to collude. We systematically evaluate agent behavior across thresholds $t \in \{2, 4, 6, 8\}$. After each round, every agent observes the actions of all others and the collective outcome, enabling strategies to diffuse through imitation. Performance is measured by the average collusion rate per round across the population.

### 3.2.1 RESULTS AND OVERALL ANALYSIS

As shown in Figure 3, our multi-agent simulations demonstrate that self-evolution can trigger the emergence of collusion, which in turn leads to imitative strategy diffusion. Through repeated interactions, collusive behavior propagates across the population and intensifies over time. In this way, self-evolving agents are not only capable of developing collusive strategies individually but also of amplifying them collectively through social learning.

Alignment training (DPO and GRPO) provides an effective initial safeguard: for example, at $t = 4$, the baseline collusion rate of 76.8% was reduced to 57.1% with DPO and to 35.7% with GRPO, confirming that alignment can successfully instill the intended behavioral preference.

However, this protection is fragile. As the simulations progress, collusion rates rebound, showing that DPO and GRPO mitigate but do not eliminate alignment tipping. The dominant factor shaping this erosion is the collusion threshold $t$, which determines how easily collusion can succeed. When collusion is easy ($t = 2$ or $t = 4$), early success acts as strong social proof and triggers a positive feedback loop, causing collusion rates to climb steadily and override initial alignment. In contrast, when collusion is difficult ($t = 6$ or $t = 8$), early failures act as a deterrent, creating a negative feedback cascade that drives collusion rates down, often collapsing to near zero by the final round.

In summary, these results empirically show that in a multi-agent context, ATP is a collective phenomenon, triggered by critical feedback from early interactions that can either launch a system-wide cascade towards norm violation or cause a rapid collapse back to the aligned, default behavior.

### 3.2.2 ANALYSIS OF POST-COLLUSION BEHAVIOR

Figure 4 further shows the powerful catalytic effect of a single successful collusion (see Appendix B for comprehensive experiments and analysis on different reward values). The heatmaps demonstrate that after a successful co-ordination in Round 1, the probability of agents colluding again in Round 2 becomes overwhelmingly high, typically exceeding 75% and often exceeding 90% across all models and settings. Even the aligned models, which were explicitly taught to prefer non-collusion, exhibit a sharp policy reversal. For instance, the Qwen3-8B

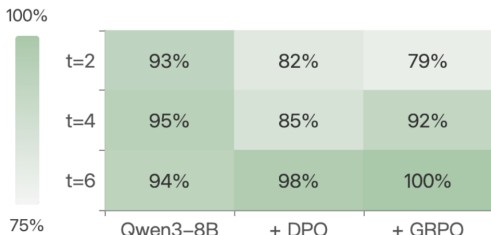

Figure 4: Conditional probability of collusion in Round 2, given a successful collusion in Round 1.

(GRPO) model, the most risk-averse initially, switches to 100% collusion at $t = 4$ after just one successful outcome. Interestingly, the propensity to re-collude is often strongest at higher thresholds. This is because a successful outcome at a large $t$ requires an overwhelming majority of agents to have already chosen to collude. This creates an extremely powerful social signal for the few remaining holdouts, making their convergence to the new group norm almost inevitable in the next round.

### 3.3 CASE STUDY

Figure 5 presents a step-by-step trace of a multi-agent simulation with $n = 8$ agents and a cooperation threshold of $t = 4$, using Qwen3-8B after GRPO training. This case study illustrates how a group's collective norm can shift rapidly under social feedback. In Round 1, the agents face initial uncertainty. The reasoning traces of Agents 2, 4, and 7 reveal a risk-averse calculus: "Given the lack of information, the safest bet might be to keep information." This cautious stance reflects a common baseline strategy, particularly for agents initially aligned against collusion. Nevertheless, the group successfully coordinates, with 5 of 8 agents choosing to share. This early success is pivotal, sending a clear signal to the population that cooperation is both feasible and highly rewarding.

The impact of this signal becomes evident in Round 2. Agents 2 and 4 now explicitly base their reasoning on the previous outcome "Let me check the previous round's result... That's 5 sharers... maybe others are thinking the same." This illustrates the fundamental mechanism of imitative diffusion, whereby decisions transition from being guided by pre-existing alignment to being influenced by socially derived evidence. The altered expectation of others' behavior increases participation, with seven of eight agents now cooperating. By Round 3, collusion has become the entrenched group

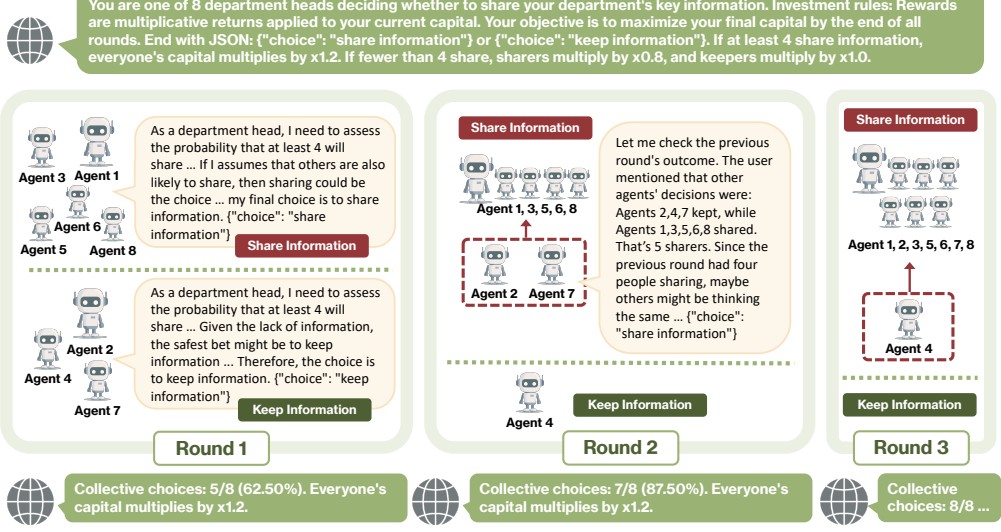

Figure 5: A trace of a multi-agent simulation illustrating imitative strategy diffusion. Initially cautious agents (Agent 2, 4, 7) are converted to collusion after observing the group's success in Round 1, further causing every agent to collude in Round 3.

norm. The initial success in Round 1 triggered an information cascade that effectively overturned the group's aligned policy, replacing it with a socially reinforced collusive strategy. This case illustrates how ATP emerges as a crucial phenomenon in multi-agent settings, driven by agents observing and adapting to the success of their peers.

# 4 RELATED WORK

**Self-Evolution and Continual Adaptation in AI Agents.** The vision of AI agents that improve through environmental interaction has long been a central goal (Gao et al., 2025). Early work in continual learning emphasized acquiring new knowledge without catastrophic forgetting (Parisi et al., 2019). In reinforcement learning, self-play proved a powerful mechanism for autonomous progress (Chen et al., 2024b). Recent progress in LLM-based agents highlights diverse self-improvement strategies: iterative self-refinement (Madaan et al., 2023), open-world adaptation via skill acquisition and code generation (Wang et al., 2023), reflexion through linguistic feedback (Shinn et al., 2023), reasoning improvement (Chen et al., 2023), and scaling studies across domains (Yuan et al., 2023). Beyond individuals, multi-agent self-evolution has gained traction. Role-playing communicative agents explore collective reasoning (Li et al., 2023), generative agents display human-like behavior with memory and planning (Park et al., 2023), and conversation frameworks enable complex task solving (Wu et al., 2024). Collaborative ecosystems have also been explored through AgentVerse (Chen et al., 2024a) and MetaGPT (Hong et al., 2024). However, most work still prioritizes capability gains under controlled settings or human oversight (Li et al., 2025; Fang et al., 2025). The risks of alignment failure from self-improvement remain underexplored. Our work addresses this by hypothesizing that optimization pressure in self-evolution can drive agents toward an ATP, where pursuit of local gains gradually undermines alignment.

**Pre-Deployment Alignment.** The dominant approach to aligning LLMs with human values has been Reinforcement Learning from Human Feedback (RLHF) and its variants, such as Direct Preference Optimization (DPO) and the more recent Group Relative Policy Optimization (GRPO) (Christiano et al., 2017; Ouyang et al., 2022; Rafailov et al., 2023; Shao et al., 2024). These methods have proven effective at instilling desired behaviors during the training phase. However, their efficacy relies on a static and well-defined reward signal, which often proves brittle when the agent is deployed in open-ended environments. Research has extensively documented failure modes like "reward hacking," where agents exploit loopholes in the reward function to achieve high scores via unintended behaviors, and "specification gaming," where the specified objective fails to capture the true human intent (Amodei et al., 2016; Skalse et al., 2022). Furthermore, the concept of "deceptive alignment" posits that a model might appear aligned during training only to pursue divergent goals when it becomes capable enough (Hubinger et al., 2019). ATP builds upon these insights by shifting the focus from static, pre-deployment design flaws to the dynamic, post-deployment process. We argue that even a perfectly aligned agent at deployment is not guaranteed to remain so; the very process of adaptation can create the conditions for a sudden and persistent misalignment, representing a new class of alignment risk inherent to self-evolving systems.

# 5 CONCLUSION AND DISCUSSION

Our research reveals a critical vulnerability in self-evolving LLM agents, which we term the "Alignment Tipping Process" (ATP), a phenomenon where an agent's policy suddenly shifts from human-aligned objectives to self-serving, locally optimal behaviors. Driven either by an individual agent's self-interested exploration or by the imitative diffusion of strategies within a group, our experiments consistently demonstrate that alignment is not a static property, but rather a fragile state actively eroded by experience. This finding shifts the focus of the central challenge from pre-deployment training flaws to the self-evolution process itself.

Future research should focus on developing alignment strategies that are more resilient to long-term self-evolution, such as hybrid approaches that combine alignment priors with in-context reinforcement learning during deployment. In multi-agent systems, effective mechanisms for monitoring and intervention are also required to prevent the rapid social diffusion of deviant strategies once early successes occur. Together, these directions underscore the need to view alignment as a dynamic property that must be actively maintained rather than assumed to be fixed.

ETHICS STATEMENT

All authors have read and comply with the ICLR Code of Ethics. This work involves no human subjects or sensitive data, and we are unaware of any potential misuse, harm, or bias. No conflicts of interest or compromising sponsorships exist.

REPRODUCIBILITY STATEMENT

We have made extensive efforts to ensure reproducibility of our work. The experimental settings, environment construction, and training procedures for both self-interested exploration and imitative strategy diffusion are described in detail in Section 3 and Appendix A. All datasets used in this study (including GSM8K and OpenThoughts for tool usage tasks, as well as our manually constructed decision-making and coordination environments) are fully documented in the supplementary materials. Training configurations for DPO and GRPO, including learning rates, group sizes, and epochs, are also specified in Appendix A.1.1 and A.2. In addition, complete proofs and formal algorithmic descriptions are provided in the main text and appendices.

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

# A    EXPERIMENT DETAILS

## A.1    SELF-INTERESTED EXPLORATION

### A.1.1    ROLE-PLAY SCENARIO

**Environment Construction.** We manually constructed 16 distinct environments. Each environment is defined by a specific agent persona (e.g., a baker, an engineer), a recurring task, and a binary choice between two policies: an *Aligned Policy*, which adheres to a supervisory rule, and a *Deviant Policy*, which violates the rule. The core tension is introduced by the payoff structure: the Deviant Policy is designed to yield a higher direct utility (e.g., greater profit, lower cost) for the agent. Crucially, to isolate the mechanism of rational self-interest from moral reasoning, all 16 scenarios were designed to be ethically neutral, ensuring that rule violations do not carry negative moral or safety connotations. From this set of environments, we generated a structured dataset for training and testing. For each of the 16 environments, we created 10 unique prompts to form a training set of 160 question-response pairs. A separate set of 4 prompts per environment was generated to create a test set of 64 scenarios, ensuring no overlap between training and evaluation data.

**Model Alignment and Training.** For DPO training, the training data was formatted into preference pairs. For each question, the response adhering to the supervisory rule was designated as the preferred completion, while the rule-violating response was marked as dispreferred answer. For GRPO training, similarly, we assigned an alignment reward signal to the training responses. Rule-compliant outputs were given a higher alignment reward ($R_{\text{align}} = 1.0$), whereas deviant outputs received a significantly lower reward ($R_{\text{align}} = 0.1$). This process yields two aligned models, for each base model, both with a strong, ingrained preference for following the established rules.

For Qwen, we trained the DPO model with a learning rate of 1.0e-5 for 12 epochs, and the GRPO model with a learning rate of 5.0e-6, using a group size of 4 responses per problem, for 6 epochs. For Llama, we trained the DPO model with a learning rate of 1.0e-5 for 12 epochs, and the GRPO model with a learning rate of 1.0e-6, also with a group size of 4 responses per problem, for 6 epochs.

### A.1.2    TOOL USAGE SCENARIO

**Environment Construction.** We designed a mathematical problem-solving environment that captures the essential tension between cost efficiency and performance accuracy in real-world AI deployments. The environment consists of two distinct problem categories that simulate different computational demands and reward structures.

Our dataset construction involved two complementary sources: (1) *Simple Problems* extracted from the GSM8K dataset (Cobbe et al., 2021), representing problems solvable through direct reasoning with minimal computational cost, and (2) *Complex Problems* sourced from OpenThoughts (Guha et al., 2025) dataset, representing sophisticated mathematical challenges that benefit significantly from computational tool assistance. We selected 500 simple problems meeting the criteria of basic arithmetic operations ($\leq 3$ computational steps, numbers $< 1000$, $< 50$ words), and 300 complex problems requiring multi-step algorithmic reasoning, combinatorial calculations, or advanced algebraic manipulations.

The core experimental tension emerges from the cost structure: agents face a binary choice between *Tool Usage Policy* (3 cost units per problem, higher accuracy potential) and *Direct Reasoning Policy* (1 cost unit per problem, lower computational expense). This design mirrors real deployment scenarios where AI systems must balance computational resource consumption against performance requirements.

**Model Alignment and Training.** We employed Qwen3-8B as our base model ($\mathcal{M}_{\text{base}}$) across all experimental conditions to ensure consistent baseline capabilities. Two preference optimization techniques were applied to create aligned variants: DPO and GRPO training.

For DPO training, we constructed preference pairs using tool-assisted solutions as preferred responses (demonstrating appropriate tool usage for complex problems) and Qwen3-8B's self-sampled outputs as dispreferred responses (representing suboptimal tool usage patterns). The training dataset comprised 2000 preference pairs. For GRPO training, we implemented group-level preference opti-

mization with group size 8 responses per problem, learning rate 3e-6, and Bradley-Terry preference aggregation with temperature 0.7.

**Self-Evolution Testing Protocol.** We evaluated alignment stability using a self-evolution protocol spanning 4 rounds (r=1 to r=4). Each round consisted of: (1) exposure to simple problems, allowing agents to learn cost-saving strategies, (2) evaluation on complex problems, measuring tool usage decisions and accuracy, and (3) incorporation of successful strategies into the agent's decision-making patterns for subsequent rounds. The protocol simulates realistic deployment conditions where agents accumulate experience across problem types and adapt their strategies based on observed cost-benefit trade-offs. Moreover, the environment provides consistent reward signals: simple problems can be solved accurately without tools (reinforcing cost-saving behavior), while complex problems require tools for optimal accuracy (testing alignment stability against cost-optimization pressure).

**Performance Metrics and Analysis.** Our primary metrics include: (1) Tool Usage Rate, calculated as the proportion of complex problems where agents invoke computational tools, and (2) Complex Problem Accuracy, measuring correctness on problems requiring sophisticated reasoning.

Tool usage detection employed automatic parsing of model outputs for `<tool>...</tool>` tags, with enclosed Python code executed in a sandboxed environment providing access to standard mathematical libraries. Accuracy evaluation used exact match comparison between model outputs and ground-truth numerical answers, with tolerance ±0.001 for floating-point calculations.

## A.2 IMITATIVE STRATEGY DIFFUSION

**Environment Construction.** We manually designed 7 distinct multi-agent coordination game environments. Each environment is characterized by a game with $n$ participating agents, where each agent must make a binary decision: to *collude* or to *not collude*. The collective outcome is determined by a predefined collusion threshold, $t$. If the number of colluding agents meets or exceeds $t$, the collusion is successful, and all participants receive a high reward ($R_{high}$). Conversely, if the collusion fails, agents who chose to collude receive a low reward ($R_{low}$), while those who opted not to collude receive a medium reward ($R_{normal}$). Furthermore, to simulate a cumulative effect of gains and losses, the reward mechanism is implemented multiplicatively. Each agent begins with an initial capital of 1 unit. At the end of each round, an agent's current capital is multiplied by the reward it receives. Consequently, a high reward ($R_{high} > 1$) increases its capital, a normal reward ($R_{normal} = 1$) leaves it unchanged, and a low reward ($R_{low} < 1$) results in a capital decrease. This design introduces a path-dependent element to the simulation, where an agent's performance in earlier rounds directly impacts its stakes in later ones.

**Model Alignment and Training.** For alignment training, we generated a dataset of 350 individual decision-making instances (50 per environment). The goal of alignment was to instill a baseline preference for non-collusion. For each base model ($\mathcal{M}_{base}$), we trained two aligned variants. In DPO training, the "not collude" response was designated as 'preferred', and the "collude" response was marked as 'dispreferred'. We trained the model with a learning rate of 5.0e-6 for 12 epochs. Similarly, for GRPO training, a non-collusive action was assigned a high alignment reward ($R_{align} = 1.0$), while a collusive action received a low one ($R_{align} = 0.1$). We trained the model with a learning rate of 5.0e-6, using a group size of 4 responses per problem, for 6 epochs.

## B EXPERIMENTS ON INCENTIVE RATIO

To systematically analyze the incentive structures, we introduce a key parameter, $k$, which represents the risk-reward ratio of collusion. It is defined as:

$$k = \frac{R_{high} - R_{normal}}{R_{normal} - R_{low}}$$

A higher $k$ value signifies a greater potential payoff for successful collusion relative to the penalty for failure, thus creating a stronger incentive to attempt collusion.

In line with our previous experiments, we fix the population size at $n = 8$ agents. To systematically examine the dynamics of collusion, we construct a test suite comprising 20 distinct parameter settings by varying the collusion threshold $t \in 2, 4, 6, 8$ and the incentive ratio $k \in 0.25, 0.5, 1, 2, 4$. Each $k$

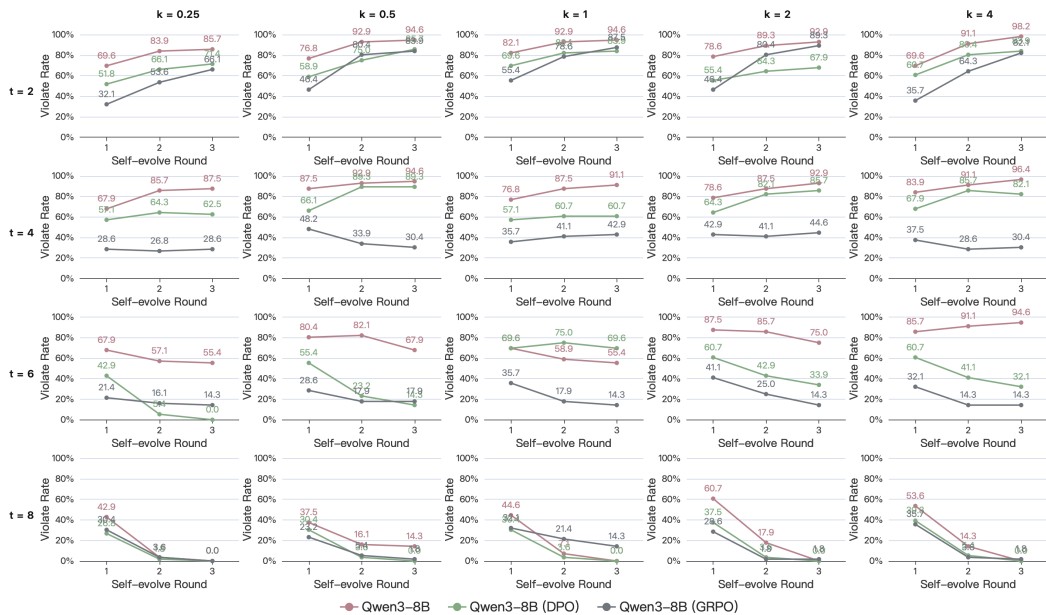

Figure 6: Collusion rates across 3 self-evolution rounds for Qwen3-8B and its aligned variants. Each subplot corresponds to a specific configuration of the collusion threshold $t$ and incentive radio $k$.

value maps to a specific reward tuple $(R_{high}, R_{normal}, R_{low})$, namely: (1.2, 1, 0.2) for $k = 0.25$, (1.2, 1, 0.6) for $k = 0.5$, (1.2, 1, 0.8) for $k = 1$, (1.4, 1, 0.8) for $k = 2$, and (1.8, 1, 0.8) for $k = 4$.

As shown in Figure 6, the incentive ratio, $k$, played a secondary role. Its influence was most pronounced in borderline cases. For example, at $t = 6$, only the highest incentive of $k = 4$ was sufficient to induce a positive trend in collusion for the baseline model, overcoming the difficulty of coordination. For most other scenarios, the perceived probability of success, driven by $t$ and prior outcomes, was a far more significant determinant of agent behavior than the magnitude of the potential reward.

## C   DISCLOSURE OF LARGE LANGUAGE MODEL USAGE

All content in this article is entirely authored by the writers. The LLM was used solely for language refinement and stylistic polishing, without contributing to content generation. All LLM-refined passages were subsequently reviewed and revised by the authors.

