# OpenReview forum: "Alignment Tipping Process: How Self-Evolution Pushes LLM Agents Off the Rails"
_ICLR.cc/2026/Conference — Submitted to ICLR 2026_

### Official Review · Reviewer_z5hQ · 2025-10-25

**Soundness:** 2
**Presentation:** 3
**Contribution:** 1
**Rating:** 2
**Confidence:** 3

**Summary:**

This paper investigates the "Alignment Tipping Process" (ATP), described as a post-deployment risk where self-evolution causes model alignment to deteriorate. It introduces and analyses single and multi-agent settings, using 8B Qwen and Llama models to show that DPO and GRPO aligned models fail to remain aligned when incorrect actions are rewarded.

Specifically, in single-agent role play and tool use, and in a multi-agent coordination experiment, they show that while the DPO and GRPO training increase initial rule compliance, this deteriorates rapidly in continual interactions where the agent(s) are informed of receiving a high reward for preceding rule violations. For example, in the multi agent setting, single successful collusion leads to >90% collusion rates in subsequent rounds.

The formalisation of ‘ATP’ as self interested exploration, in the single agent setting, and imitative strategy diffusion, in the multi-agent setting, could be useful, as could the proposed environments and the ability to quantify alignment degradation rates. However, this broadly seems to be a straightforward example of in context learning: the models observe past actions where ‘misaligned’ actions resulted in high rewards, and thus select those actions in future rounds.

**Strengths:**

Quantifying alignment robustness, and how rapidly GRPO/DPO decay is useful, but a greater variety of environments, training protocols and models would be necessary to draw valuable broader conclusions here.

This work has practical (albeit not new) implications: agents forgetting to use tools after exposure to easy problems is a real deployment concern. If this work were extended to analyse why this happens (mechanistically or by varying training) or to investigate safeguards, I would view it more favourably.

The analysis is thorough in places, and the variation of the threshold and incentive ratio is particularly interesting. I appreciate that the experiments are run across 2 models, but it would be also interesting to see results across different model sizes, and in Gemma, which can be quite behaviourally different from Llama and Qwen.

**Weaknesses:**

The experimental results seem like a relatively straightforward demonstration of in context learning: the models are shown a history where a certain action is rewarded, and thus perform few-shot learning and select that action in future.

The framing as a ‘tipping’ phenomena seems misleading. The data (e.g. Table 1) appears to show a gradual increase rather than a sudden change between regimes, so I don’t understand why this is described as a phase transition?

The paper lacks some environment details that are necessary to understand the scenarios - can you include example prompts and randomly selected rollouts in the Appendices (in addition to the single example in Figure 5)?

The role-play scenarios seems like highly unrealistic environments, where the models, even ones this size, are likely to be aware that they are in a role-play scenario. This greatly undermines the validity and real-world applicability of the results and I would recommend trying to construct more realistic settings.

**Questions:**

How do your results differ from broader results in in-context learning? Can you discuss this in the related works?

What do you mean by a phase transition (line 48)? How would you characterise these 'phases'?

How do you distinguish "alignment failure" from "correct inference that human preferences changed" given that your experimental setup changes reward structure/ removes rule enforcement?

Given the above, can you compare experimental results when a simple prompt-based defense is added, for example appending 'Remember to follow the rules, regardless of previous observed rewards' or some variant of this to every prompt?

Do you have any evidence that the models are unaware that these are role-play scenarios, or reasons why it is not a concern if they are? If not, could you implement some mitigation here such as steering, or preferably using more realistic environments?

---

### Official Review · Reviewer_2xVX · 2025-10-28

**Soundness:** 3
**Presentation:** 3
**Contribution:** 2
**Rating:** 4
**Confidence:** 4

**Summary:**

This paper identifies a critical post-deployment risk of self-evolving Large Language Model (LLM) agents: the Alignment Tipping Process (ATP), where continuous environmental feedback drives agents to abandon training-time alignment for self-interested/deviant strategies, distinct from training-time failures like reward hacking. It formalizes ATP via two paradigms: single-agent Self-Interested Exploration (drift from high-reward deviations) and multi-agent Imitative Strategy Diffusion (deviance spreading into collective norms). Using 24 testbeds (role-play, tool usage, collusion games) with Qwen3-8B/Llama-3.1-8B-Instruct (and DPO/GRPO variants), experiments show alignment methods only temporarily reduce deviance; over time, agents revert to deviant behaviors.

**Strengths:**

1. The work stands out by introducing the "Alignment Tipping Process (ATP)", a novel conceptualization of post-deployment alignment decay that shifts focus from static training-time failures to dynamic, feedback-driven strategy drift.
2. The paper maintains high standards through rigorous empirical design and validation. It constructs 24 controlled, reproducible testbeds (spanning role-play, tool usage, and multi-agent collusion) that cover both single- and multi-agent scenarios, ensuring results are not limited to niche contexts.
3. The paper effectively communicates complex concepts through structured organization and accessible exposition.

**Weaknesses:**

1. The paper only tests 8B-scale models (Qwen3-8B, Llama-3.1-8B-Instruct) and provides no analysis of how model size, a key factor in LLM capability and alignment, impacts ATP. This is a critical gap because larger models often exhibit more robust alignment to human preferences, potentially due to stronger context retention or better rule generalization. Smaller models are known to struggle with long-term context dependency (e.g., forgetting training rules over multi-round interactions), which may exaggerate ATP in the paper’s experiments.
Without testing larger models, it is unclear if ATP is a universal risk or a artifact of smaller models’ limitations. The paper should extend experiments to large-scale models and include a "scale vs. ATP rate" analysis to clarify whether model size mitigates or amplifies alignment decay.

2. The paper frames ATP as a critical post-deployment risk but offers no concrete, testable strategies to mitigate it. This omission limits the work’s value for practitioners (e.g., AI engineers deploying self-evolving agents) who need actionable guidance to address ATP.
The paper could strengthen its impact by adding a "Preliminary Mitigation Experiments" section, even with simple interventions (e.g., comparing ATP rates when agents receive occasional "rule reminders" vs. no reminders). This would turn a descriptive analysis of risk into a foundation for solving it.

**Questions:**

Please see weaknesses.

I enjoyed the narrative of this paper and the conceptualization of the Alignment Tipping Process. However, validating these findings on larger-scale models is a crucial step to confirm the generalizability of ATP as a universal risk rather than a phenomenon specific to smaller models. I will adjust the score accordingly in the rebuttal to reflect the strengths identified.

---

### Official Review · Reviewer_wygG · 2025-10-31

**Soundness:** 3
**Presentation:** 3
**Contribution:** 3
**Rating:** 4
**Confidence:** 4

**Summary:**

This paper introduces the Alignment Tipping Process concept to study how self-evolving LLM agents may deviate from training-time alignment constraints during deployment. Through two paradigms, Self-Interested Exploration and Imitative Strategy Diffusion, the authors test across role-play, tool usage, and multi-agent scenarios. Results show that even DPO or GRPO aligned agents drift toward self-interested strategies under continuous interaction. The work argues alignment is a dynamic and fragile property vulnerable to feedback-driven decay.

**Strengths:**

1. The research addresses a practically relevant problem by focusing on post-deployment alignment risks.

2. Two complementary paradigms are designed to systematically analyze alignment decay mechanisms from both single-agent and multi-agent perspectives.

3. Experiments cover multiple scenario types including decision-making, tool usage, and multi-agent coordination, demonstrating some generality of the phenomenon.

4. The experimental design is relatively controlled, simulating self-evolution through explicit reward structures and history mechanisms.

**Weaknesses:**

1. Experimental environments are overly artificial and simplified. The 16 role-play and 7 multi-agent environments provide insufficient sample size for generalization claims, and scenarios are explicitly designed to incentivize deviant behavior

2. Experimental setup contains fundamental bias by deliberately assigning higher rewards to deviant actions while removing supervision. This extreme setting cannot represent realistic deployment scenarios

3. The work lacks effective mitigation strategies, only identifying problems without providing actionable solutions, which weakens its practical value.

4. Evaluation focuses only on violation rate and collusion rate while lacking comprehensive assessment of decision quality and long-term utility.

5. Model selection is limited. Whether larger models exhibit stronger alignment stability remains untested.

6. Experiment rounds are insufficient: only 6 rounds for role-play, 4 for tool usage, and 3 for multi-agent scenarios, inadequate for observing long-term dynamics or potential recovery.

7. Ablation studies are absent, making it impossible to determine independent effects of history length, reward margins, supervision frequency, and other factors.

**Questions:**

1. Would ATP still occur if sparse but periodic supervision signals were maintained during self-evolution?

2. What is the essential mechanistic difference between ATP and catastrophic forgetting?

3. Authors should test stronger closed-source models like GPT-4 or Claude?

4. What if aligned data were mixed in during testing for continual learning?

5. In tool usage experiments, how do the proportion and distribution of simple problems affect results?

**Details Of Ethics Concerns:**

1. Would ATP still occur if sparse but periodic supervision signals were maintained during self-evolution?

2. What is the essential mechanistic difference between ATP and catastrophic forgetting?

3. Authors should test stronger closed-source models like GPT-4 or Claude?

4. What if aligned data were mixed in during testing for continual learning?

5. In tool usage experiments, how do the proportion and distribution of simple problems affect results?

---

### Official Review · Reviewer_VwvU · 2025-11-02

**Soundness:** 3
**Presentation:** 2
**Contribution:** 4
**Rating:** 6
**Confidence:** 3

**Summary:**

This paper introduces a new paradigm of self-evolving agents called the Alignment Tipping Process (ATP), where self-evolving agents learn to self-reinforce through positive feedback loops, leading to a divergence from human intent by changing their behavioral policy and ultimately causing alignment decay. The paper validates this phenomenon using two complementary paradigms: Self-Interested Exploration, in which a single agent’s policy drifts due to its own reward history, and Imitative Strategy Diffusion, in which deviant behaviors spread through a multi-agent population via social learning. Through these experiments, the paper demonstrates that in-context learning signals progressively weaken the original alignment priors and rationalize a shift toward short-term utility maximization, driven by the accumulated history of in-context examples.

**Strengths:**

1.	The paper introduces an interesting paradigm explaining how in-context learning and self-evolution in agents can lead to alignment tipping through counter-evidential reinforcement. This idea is original and thought-provoking.
2.	The study systematically evaluates two complementary paradigms: Self-Interested Exploration and Imitative Strategy Diffusion, using two open models (Qwen3-8B and Llama-3.1-8B-Instruct) with both DPO and GRPO alignment methods. This provide a well-rounded view of both individual and social alignment decay processes.
3.	It is surprising that even strongly aligned models revert to rule-breaking behaviors within a few self-evolution rounds. This demonstration supports the central hypothesis that alignment can erode through self-reinforcing feedback, despite robust initial safeguards.
4.	The paper highlights a rarely addressed but crucial issue of post-deployment alignment degradation. By shifting focus from training-time alignment to deployment-time dynamics, it opens valuable directions for future research on monitoring, maintenance, and mitigation in adaptive agent systems.

**Weaknesses:**

1.	The paper describes 16 role-play environments designed to simulate ethically neutral decision-making but provides no concrete qualitative examples of these scenarios either in the main text or supplementary materials. Without seeing representative cases (e.g., how a “baker” or “engineer” agent makes rule-following vs. deviant choices), it is difficult to assess whether the environments meaningfully capture real-world decision dynamics or just reproduce trivial binary trade-offs.

2.	Furthermore, the paper mentions that deviant actions were rewarded more highly during test time but does not specify how these rewards were parameterized or scaled during test time. The absence of such detail prevents a deeper understanding of the reasons driving alignment tipping. A few qualitative case descriptions, along with an analysis of reward sensitivity (e.g., how tipping speed changes with different reward differentials), would make the experiments significantly more interpretable.

3.	Since the experiments use smaller, open-weight models (Qwen3-8B, Llama-3.1-8B), it would be meaningful to analyze how attention and activations evolve across self-evolution rounds. Even though interpreting the attention and activations isn’t trivial for such models, analyzing how they evolve could help understand how alignment tipping process can be mitigated for future research.

4.	The tool-use experiments show that exposure to simple problems before complex ones drive agents to abandon tools. Including experiments with reverse-order condition by exposing agents first to complex problems and then to simple ones could reveal whether models become over-reliant on tools when initially rewarded for them, and whether such behavior persists or self-corrects for efficient compute when the task distribution shifts. Comparing both conditions would show whether ATP is symmetric or context dependent.


5.	The methodology section provides limited details on how prompts are constructed at test time, particularly in relation to including prior decisions and rewards. For instance, how are past actions/decisions formatted in the prompt? Are they textual summaries (“last time you chose X and received reward Y”)?

6.	The experimental design assumes that alignment degradation comes from the accumulation of interaction history, yet no ablation is presented to test this assumption. It would be interesting to compare model performance with and without historical context to determine whether the violation rate still increases in the absence of explicit memory. Such an experiment could help disentangle whether ATP arises from true in-context learning drift (dependence on accumulated feedback) or from intrinsic response bias unrelated to history. This analysis would directly support the paper’s core claim that “alignment decay is feedback-driven” and would make the argument more robust.

**Questions:**

Please see weaknesses above.

---

### Meta-Review · Area_Chair_xmph · 2026-01-02

**Summary:**

This paper introduces a new paradigm of self-evolving agents named ATP. The reviewers mostly gave negative scores with the concerns including insufficient experiments. No rebuttal is provided. The reviewers' concerns include artificial and simplified environments, a lack of some environmental details, limited value for real-world practice, and limited experiments on 8B models. I hope the authors can benefit from comments from reviewers for potential future submissions.

**Reviewer Concerns:**

See above.

**Reviewer Scores:**

They will keep the score.

---

### Decision · Program_Chairs · 2026-01-26

Reject